# Alkaliphilic/Alkali-Tolerant Fungi: Molecular, Biochemical, and Biotechnological Aspects

**DOI:** 10.3390/jof9060652

**Published:** 2023-06-09

**Authors:** Maikel Gilberto Fernández-López, Ramón Alberto Batista-García, Elva Teresa Aréchiga-Carvajal

**Affiliations:** 1Unidad de Manipulación Genética, Laboratorio de Micología y Fitopatología, Facultad de Ciencias Biológicas, Universidad Autónoma de Nuevo León, San Nicolás de los Garza 66451, Mexico; 2Centro de Investigación en Dinámica Celular, Instituto de Investigación en Ciencias Básicas y Aplicadas, Universidad Autónoma del Estado de Morelos, Cuernavaca 62209, Mexico

**Keywords:** extremophile microorganisms, pH, alkaliphilic/alkali-tolerant fungi

## Abstract

Biotechnologist interest in extremophile microorganisms has increased in recent years. Alkaliphilic and alkali-tolerant fungi that resist alkaline pH are among these. Alkaline environments, both terrestrial and aquatic, can be created by nature or by human activities. *Aspergillus nidulans* and *Saccharomyces cerevisiae* are the two eukaryotic organisms whose pH-dependent gene regulation has received the most study. In both biological models, the PacC transcription factor activates the Pal/Rim pathway through two successive proteolytic mechanisms. PacC is a repressor of acid-expressed genes and an activator of alkaline-expressed genes when it is in an active state. It appears, however, that these are not the only mechanisms associated with pH adaptations in alkali-tolerant fungi. These fungi produce enzymes that are resistant to harsh conditions, i.e., alkaline pH, and can be used in technological processes, such as in the textile, paper, detergent, food, pharmaceutical, and leather tanning industries, as well as in bioremediation of pollutants. Consequently, it is essential to understand how these fungi maintain intracellular homeostasis and the signaling pathways that activate the physiological mechanisms of alkali resistance in fungi.

## 1. Introduction

Biotechnological processes require the use of products mainly from mesophilic microorganisms, which thrive well in environmental conditions close to those supported by human cells [1]. However, some organisms do not operate under these conditions and are adapted to extreme environments. Microorganisms that develop and grow under unusual environmental conditions are referred to as extremophiles, a term coined by Macelroy in 1974 [2]. Extremophiles are microorganisms that can grow in extreme physical or geochemical conditions [3], including high and low temperatures, extreme pH or salinity, presence of radiation, extreme pressure, or very high hydrocarbon concentration [4].

Despite the general preference of fungi to grow at neutral or slightly acidic pH, it has been demonstrated that some can also thrive at high pH [5]. These organisms produce alkaline enzymes capable of resisting high pH (proteases, amylases, and lipases) and can be used in the textile, food, pharmaceutical industry, etc. [6,7,8]. Even today, the terms alkaliphilic and alkali-tolerant can be perplexing because this classification is highly dependent on the growth conditions and culture medium composition. Horikoshi (1999) [9] postulated that alkaliphiles are microorganisms that grow optimally or exceptionally well at pH values above 9, typically between 10 and 12, but cannot grow or grow only slowly at the near-neutral pH value of 6.5. Grum-Grzhimaylo et al. (2016) [5] proposed an even more comprehensive classification; they define alkaliphiles as organisms that thrive at pH levels greater than 8. These alkaliphiles can be either obligate (they cannot grow at pH 4–5) or facultative (they can grow at acidic pH). While alkali-tolerant fungi can grow at alkaline pH, their optimal growth occurs at pH levels below pH 8. Depending on their growth profile at different pH, these can be further classified as strongly, moderately, or weakly alkali-tolerant. According to the definition proposed by Grum-Grzhimaylo et al. (2016) [5], we will use the terms alkaliphilic and alkali-tolerant in this review.

Alkaline environments, both terrestrial and aquatic, can be created by nature or by humans. Alkaline environments of anthropogenic origin can be derived from cement manufacturing, indigo dye production, mining operations, paper and pulp production, and food processing effluents, among others [10]. Ca^2+^ and Mg^2+^ are indeed initially scarce in natural environments; therefore, the dissolution of atmospheric CO_2_ coupled with the rapid evaporation of water results in the formation of high pH environments up to pH 12 [1], leaving Na^+^ as the dominant cation [11]. In calcium-containing systems, however, the accumulated carbonate anion is swiftly involved in the formation of insoluble calcite, CaCO_3_, and dolomite, MgCa(CO_3_)_2_, resulting in an environment that is nearly pH neutral [1]. Nevertheless, each of these ecosystems has its own physicochemical characteristics that impose a unique biodiversity, which is of interest to study fungal adaptative evolution of (poly)extremophilic microorganisms, i.e., alkaliphilic and alkali-tolerant fungi, as well as for its characteristics as a unique source for biomolecules with biotechnological potential.

The pH-dependent signaling system involving the transcription factor PacC/Rim101, which is proteolytically activated by the Pal/Rim signaling pathway, is one of the most thoroughly investigated alkali gene-regulating strategies to modulate metabolism in fungi when grown in alkaline conditions. PacC/Rim101 transcription factor activation activates genes related to alkaline stress and inhibits genes related to growth in the presence of acidic pH [12]. This review aims to synthesize information regarding alkaliphilic and alkali-tolerant fungi, their adaptation mechanisms, related signaling cascades, and the biotechnological significance of the enzymes that can be produced under this stress.

## 2. Fungal Diversity in Alkaline Environments

It is recognized that abiotic factors inhibit the development of most organisms exposed to extreme habitats. Soda soils (or alkaline soils) with pH values higher than eight are an example of such extreme environments [5]. It could be assumed that the present biota in extreme ecosystems is limited, however, numerous studies have demonstrated otherwise; not only bacterial species but also eukaryotic species of various taxonomic levels have been found [5,13,14,15,16].

In the case of fungi, it is difficult to isolate alkaliphilic and alkali-tolerant strains because, in many instances, neutral or acidic isolation media are used to study the fungal diversity of alkaline soils, thereby decreasing the likelihood of recovering alkaliphilic strains [5]. Steiman et al. (2004) [17] isolated a broad variety of fungi from samples of soil and sediment with pH ranging from 8 to 9.2, which were obtained from Mono Lake (USA). The authors used Malt Extract Agar with a pH = 8 for the isolation, but no buffer to maintain the pH of the culture medium was added. This is not recommended for the isolation of alkaliphilic and/or alkali-tolerant fungi since their metabolic processes can modify the pH of the culture medium. Many of the isolates obtained from Mono Lake are likely alkali-tolerant for the aforementioned reasons. Nonetheless, the environmental conditions present in Mono Lake (USA), such as high alkalinity and salinity, are pertinent to isolating unique phenotypes of alkaliphilic fungi.

When isolating fungi from alkaline environments, the fungal identification was an important methodological challenge for early mycologists, as this was accomplished by analyzing the morphophysiological characteristics of the isolated fungi. The advent of massive sequencing made it feasible to more efficiently classify the different taxonomic groups and reclassify many of the initially isolated species. Many of the identified species in Mono Lake (USA), Iwate Prefecture (Japan), and Jayapura (Indonesia) have been reclassified into other taxonomic groups. Figure 1 shows the biodiversity of fungi in certain alkaline environments.

Despite the fact that alkaline environments inhibit the development of the majority of known mesophilic organisms, diversity bioprospection of these ecosystems has revealed a wide variety of fungi (Figure 1). Species of *Acremonium* and *Sodiomyces* have been isolated from different alkaline environments. The species of these two genera are present in fifty percent of the studied environments, and some of these species can thrive at a pH range of 10 and 11 [5]. Both genera have been isolated from soil and water samples; however, the ecosystems where *Acremonium* sp. was isolated had a pH range of 7 to 11, whereas *Sodiomyces* sp. was more exclusive of habitats with pH values above 9 due to their obligate alkaliphilicity. Other isolated species include those of the genera *Fusarium* and *Penicillium*, which were recovered from 30% to 40% of the environments depicted in Figure 1. Some of the strains from these two genera exhibited a mild to moderate alkali-tolerant profile.

On the other hand, since the sampling site in Jayapura, Indonesia, is a cultivation area, the diversity of alkaliphilic and alkali-tolerant fungi in this region is likely influenced by anthropogenic activities. Some of the fungi isolated from these ecosystems are plant pathogens (*Cercospora* sp., *Cylindrocarpon* sp., *Paramyrothecium roridum*), which is related to the sampling site’s characteristics.

The use of culture techniques has permitted the morpho-physiological study of isolated species and is an essential approach for the analysis of biodiversity. Furthermore, culture techniques are also useful to comprehend the mechanisms of adaptation to extreme pH in alkaliphilic and alkali-tolerant fungi. However, there are numerous non-cultivable fungal species whose presence can only be inferred using metagenomic tools.

Here, we also analyzed several metagenomes deposited in the MG-RAST (https://www.mg-rast.org/ (accessed on 7 May 2023)) to increase our understanding of fungal biodiversity in alkaline environments. Four natural environment metagenomes were chosen: Büdös-szék pan (Hungary), Pantanal (Brazil), Mono Lake (USA), and Lonar Lake (India). The Büdös-szék pan is a typical alkaline aquatic environment (pH = 9.16) in the Pannonian biogeographic steppe [28]. The Brazilian Pantanal biome (specifically in the Nhecolândia sub-region) hosts hundreds of pristine soda lakes with a pH gradient varying between 8.62 and 10.05 [29]. Mono Lake is an alkaline and hypersaline closed-basin lake in central California, just east of the Sierra Nevada Mountains [17]. Water samples were used in the metagenomic analysis of these three sites. The hypersaline and hyperalkaline Lonar Lake in Maharashtra, India, is a natural meteorite lake; the metagenome of this site was derived from sediment with a pH range from 9.16 to 10.16 [30].

Analyzing the relative abundance at the family level, we did not observe a significant difference between metagenomes (Figure 2). However, there are differences between the species identified in these four metagenomes and the isolated species shown in Figure 1. *Chaetomiaceae*, *Cordycipitaceae*, *Mycosphaerellaceae*, *Onygenaceae*, and *Pleosporaceae* were among the families recovered by both cultivable and metagenomic methodologies. Approximately 80% of the predicted families in the metagenomes were not recovery by cultivable methods, as shown in Figure 1. These findings demonstrate the significance of conducting biodiversity studies using metagenomic libraries to study a broader spectrum of the fungi present in these extreme environments.

## 3. Physiology

Most alkaliphiles have an optimal growth pH of around 10, which is the most significant difference from the generally investigated neutrophilic microorganisms; however, their intracellular enzymes have a maximum activity near neutral pH [31]. This evidence suggests that alkaliphilic and alkali-tolerant fungi have evolved mechanisms to overcome the effect of basic pH. Under alkali conditions, it is likely that fungi possess certain mechanisms to alleviate the influence or damage of alkali: H^+^ proton sequestration in different cellular compartments [32]. In addition, some neutrophilic fungi isolated in soils produce different organic acids to solubilize phosphate salts [33]; a similar mechanism could probably be found in alkaliphilic and alkali-tolerant fungi that contributes to the acidification or neutralization of alkaline soils [34] and morphological arrangements of membranes [35], among others.

Much of the literature that describes the mechanisms involved in alkaline pH resistance is primarily concerned with bacteria and neutrophilic fungi, but these mechanisms may also occur in alkaliphilic and alkali-tolerant fungi. In this section, we will discuss a number of these pieces of evidence and relate them to potential pH homeostasis mechanisms in alkaliphilic and alkali-tolerant fungi.

### 3.1. Genes, Transcription Factors, and Signaling Mechanism

The eukaryotic organisms in which gene regulation by pH has been most extensively investigated are the ascomycete fungi *Aspergillus nidulans* (Pal pathway) and *Saccharomyces cerevisiae* (Rim pathway) [12]. Both models are neutrophilic fungi; however, there are some reasons to use them in studies of pH variation. For instance, mutations that affect pH regulation cause phenotypic variations that are easy to identify and study in mutant lines. In the case of *A. nidulans*, the ability to detect acid and alkaline phosphatases by a simple colony staining test was decisive [36], while mutations in the Rim genes of *S. cerevisiae* cause easily detectable phenotypic changes such as poor growth at low temperatures, morphological alterations in the colonies, inefficient sporulation, and defective invasive growth [37].

pH is regulated by a signal transduction pathway that leads to the proteolytic activation of a zinc finger transcription factor called PacC [38,39] (Figure 3). PacC_72_ possesses 674 amino acids [40,41] and undergoes post-translational modifications when extracellular pH is neutral or alkaline. PacC_72_ acquires a closed conformation at acidic pH, which is maintained by the interaction of a C-terminal domain with two upstream domains, thereby preventing proteolytic processing. When the pH signal is received, the protein conformation changes to an open state [42]. PacC_72_ contains a “signaling protease box” that removes roughly 180 C-terminal residues, resulting in ∼495 residues (PacC_53_) [43]. PacC undergoes a second processing step in which an additional ∼245 C-terminal residue is removed, resulting in the ∼250 residues (PacC_27_) [40,43,44]. This second proteolysis reaction is pH-independent [43]. PacC_27_ inhibits acid-expressed genes while activating alkaline-expressed genes [45]. The PacC_27_ form exposes the DNA-binding domain (DBD) containing three Cys_2_His_2_ zinc fingers (core consensus: 5-GCCARG) [45] that is sufficient to drive preferential nuclear localization [46,47]. PacC_27_ and PacC_53_, but not PacC_72_, exhibit marked nuclear localization preference [47]. Finger 1 interacts with finger 2 instead of DNA, whereas residues within fingers 2 and 3 interact with target DNA [41,46,48].

Ascomycota’s Pal/Rim pathway comprises six gene products: Rim21/PalH, Rim8/PalF, Rim20/PalA, Rim13/PalB, Rim23/PalC, and Rim9/PalI [37,38,39,49,50,51]. The products of six genes, *palA*, *palB*, *palC*, *palF*, *palH*, and *palI*, transmit the alkaline extracellular pH signal to PacC [38,39]. Null mutations in any of the six *pal* genes of the pH signal transduction pathway and in *pacC* have an acidity-mimicking phenotype [38,39].

The pH signaling begins at a plasma membrane complex consisting of the 7-transmembrane (7TM) domain protein Rim21/PalH, the arrestin Rim8/PalF, and, most likely, the 3-transmembrane domain protein Rim9/PalI [52,53]. PalI is thought to aid in the localization of PalH to the plasma membrane, according to [52]. The C-terminal tail of PalH interacts strongly with the arrestin-like protein PalF, which contains an N-domain and a C-domain that are conserved in the arrestin family and are likely responsible for recognizing and binding to the 7TM receptor. In an alkaline pH, PalF is phosphorylated/ubiquitinated in a palH-dependent and partially PalI-dependent mechanism [53]. Arrestins play key roles in signal transduction by interacting with the cytoplasmic domains of 7TM receptors and are known to participate in endocytosis by connecting them to the endocytic internalization apparatus [54].

PalC is recruited to sites near the plasma membrane in an ambient pH- and 7TM receptor PalH-dependent manner, but its recruitment is PalA-independent. In addition, PalC binds to Endosomal Sorting Complex Required for Transport-III (ESCRT-III) Snf7p/Vps32, indicating a potential role as a link between the upstream and downstream pH signaling complexes [55]. Although multiple studies associate pH signaling proteins with endocytosis, Lucena-Agell et al. (2015) [56] demonstrated by endocytosis block that PacC processing is not significantly affected, indicating that pH signaling proteins probably do not require endocytosis.

pH signal transduction requires that the PalA protein binds to two YPXL/I motifs located at either side of the signaling protease box in PacC72, where Tyr, Pro, and Leu/Ile are crucial for its interactive properties, required for the pH-regulated cleavage of this transcription factor [57]. PalB, a cysteine protease of the calpain family, is probably the signaling protease that causes the first proteolytic processing of PacC72 but not the second proteolytic processing [58].

Data from several research groups strongly suggest that PalA/Rim20 physically interact with certain class E Vps protein complexes, acting at the cytosolic side of endosomes to mediate the sorting of transmembrane proteins into the multivesicular body pathway [57]. The endosomal trafficking pathway plays a fundamental role in the interaction of cells with their environment [59] and has critical implications on signal transduction and cell growth control. In addition, evidence suggests a connection between this pathway and pH-dependent signaling mechanisms [60].

On the other hand, the ESCRTs encompass a set of interacting protein complexes that recognize ubiquitinated membrane proteins (Ub-cargo) and facilitate their sorting into intralumenal vesicles [61]. The ESCRT-I, -II, and -III protein complexes function to create multivesicular bodies for sorting of proteins destined for the lysosome or vacuole [62]. ESCRT-0 and ESCRT-I capture cargo and initiate intralumenal vesicle formation, after which ESCRT-II and, ultimately, ESCRT-III are recruited to the ESCRT apparatus to complete vesicle formation and fission from the endosomal limiting membrane [61].

Xu et al. (2004) [60] determined whether the ESCRT components were required for the catalytic processing of Rim101, which was dependent on ESCRT-III subunits Snf7p and Vps20p but independent of Vps2p and Vps24p. Rim101 processing was also dependent on ESCRT-II subunits Snf8p (Vps22p), Vps25p, and Vps36p and on ESCRT-I subunits Vps23p, Vps28p, and Vps37p. Therefore, the majority of ESCRT subunits are required for Rim101 processing. However, it is likely that the activation of Rim/Pal genes does not require endocytosis [56]. Changes in extracellular pH alter the charge of the inner leaflet of the fungal plasma membrane, thereby modifying the biochemical interactions of the Rim21/PalH pH sensor with the cytoplasmatic membrane, for example in *S. cerevisiae* [63]. Rim8/PalF decode the alkaline pH signal received by Rim21/PalH, and PalF/Rim8 ubiquitination plays a positive role in pH signaling by recruiting ESCRTs and PalB to sites near the plasma membrane [56].

Some neutrophils have been extensively used as biological models to study pH signaling; however, these studies in alkaliphilic fungi are scarce. Using genomic tools, we can infer the presence of potential adaptations to pH if we demonstrate that alkaliphilic/alkali-tolerant fungi have the genes that encode for the Pal pathway. The genome of the alkaliphilic fungus *Sodiomyces alkalinus* was sequenced (BioSample: SAMN02745840) [64], and the UniProt annotation [65] shows that this fungus presents practically all the Pal pathway genes (PalI, PalH, PalF, PalC, PacC). Additionally, it contains sequences homologous to V-type proton ATPase. Proton-translocating ATPases (H^+^-ATPases) are rotary enzymes that couple proton (or Na^+^) translocation across membranes with ATP synthesis or hydrolysis. Vacuolar or V-type ATPases work in reverse mode by actively pumping protons through membranes using energy derived from ATP hydrolysis [66,67].

Another mechanism associated with pH homeostasis is the Na^+^/H^+^ antiporter system, which has been very well studied in bacteria, but very little is known about this response in alkaliphilic fungi. Neutralophilic bacteria utilize the proton electrochemical potential (ΔμH^+^), a sum of transmembrane pH gradient (ΔpH; inside alkaline) and membrane potential (Δψ; inside negative), in the active transport of solutes, ATP synthesis, and motility [68]. Alkaliphilic bacteria acidify the cytoplasm by a Na^+^/H^+^ antiporter system and consequently exhibit ΔpH in the opposite direction to that in neutrophiles [69]. Thus, instead of ΔμH^+^, alkaliphiles utilize a sodium electrochemical potential (ΔμNa^+^), a sum of transmembrane sodium gradient (ΔpNa; outside > inside) and Δψ, in the active transport of amino acids and motility [69,70]. Interestingly, *S. alkalinus* also has a gene that encodes an Na^+^/H^+^ exchanger. The presence of orthologs of V-type ATPases and Na^+^/H^+^ exchanger gene in *S. alkalinus* may be indicative of an adaptation mechanism in response to pH.

### 3.2. Membrane Adaptations

Glycerophospholipids and sphingolipids contribute to the plasma membrane’s asymmetry. Phosphatidylcholine (PC) and complex sphingolipids are predominantly located in the outer leaflet (extracytosolic) of the lipid bilayer. In contrast, phosphatidylserine (PS), phosphatidylethanolamine (PE), and phosphatidylinositol (PI) are restricted to the inner leaflet (cytosolic) [71,72]. The lipid translocases or flippases are responsible for moving the lipids in the membrane and establishing their asymmetry [73]. Among the translocases reported, ABC transporters catalyze flop, which is the movement from the cytosolic to extracytosolic leaflet, whereas P_4_-type ATPases stimulate flip, the reverse movement [74], and scramblases randomize the lipid distribution between the two leaflets [75]. This asymmetric structure, characterized by a greater number of anionic lipids in the cytosolic leaflet and predominantly neutral lipids in the extracellular leaflet, generates two surfaces with radically distinct electrostatic potentials [76].

There are multiple described mechanisms in fungi for maintaining intracellular homeostasis when the pH is acidic [32]. However, the relationship between plasma membrane lipid asymmetry and alkaline pH in the extracellular environment is poorly understood. Several studies demonstrate that the deletion of translocase genes in yeasts such as *Cryptococcus neoformans* and *S. cerevisiae* impairs intracellular traffic and organelle structure maintenance [77,78]. The majority of studies on membrane asymmetry and its relationship with pH are focused on *S. cerevisiae*, limiting our understanding of the effects of alkaline pH on the membrane in other taxonomic groups of fungi.

In *S. cerevisiae*, five proteins belonging to the P_4_-type ATPases family are present: Dnf1, Dnf2, Dnf3, Neo1, and Drs2 [76]. Deletion of the *dnf1* and *dnf2* genes abolishes ATP-dependent flip of fluorescent-labeled PC, PE, and PS, indicating that Dnf1 and Dnf2 have important roles in the asymmetry stability of plasma membrane glycerophospholipid [73,79]. Dnf1 and Dnf2 require association with a common subunit, Lem3 (also known as Ros3), a member of the Cdc50 family [80,81].

Many ABC transporters implicated in multidrug resistance are present in yeasts. Nevertheless, the presence of a large number of ABC transporters in yeast genomes and the fact that not all ABC members are directly involved in multidrug export indicate that they perform essential physiological functions independent of those related with the presence of drugs [82]. For instance, Pdr5 and Yor1 are believed to be implicated in glycerophospholipids flop in yeast *S. cerevisiae* [79,83]. *pdr5* and *yor1* are transcriptionally regulated by Pdr1, and their up-regulation results in a gain of function. The *pdr1-3* mutant reduces the accumulation of labeled PE, most likely owing to an increase in efflux [83]. A Pdr5/Yor1-dependent increase in endogenous PE on the cell surface was later confirmed [79].

Although glycerophospholipid translocation mechanisms has been well studied, a similar system for sphingolipid translocation has yet to be disclosed [84]. Rsb1p was identified as a putative sphingoid long-chain base-specific translocase/transporter that mediates flop or efflux of the long-chain base in an ATP-dependent manner [85] (Figure 4). Rsb1p expression may be mediated by alterations in glycerophospholipid asymmetry. In wild-type *S. cerevisiae*, under normal growth conditions in complex media, the expression level of *rsb1* is low, whereas its expression is significantly elevated if the glycerophospholipid asymmetry is disturbed. This alteration can be caused by mutations in genes involved in the flip (i.e., Dnf1, Dnf2, or Lem3) or the flop (i.e., ABC transporters Pdr5 and Yor1) of glycerophospholipids [84]. Interestingly, *rsb1* is also induced by Rim13, Rim20, and Rim21. Mutations in any of these genes decrease *rsb1* expression levels in pdr5Δ cells, indicating that Rim21, Rim20, and Rim13 regulate the *rsb1* expression via Rim101. Therefore, all mutations known to impair Rim101 activation (*rim21*Δ, *dfg16*Δ, *rim9*Δ, *rim8*Δ, *rim13*Δ, *rim20*Δ, *ygr122w*Δ, *vps28*Δ, *vps25*Δ, *snf7*Δ, and *vps20*Δ) caused *rsb1* expression reduction [73].

Phosphatidylserine is heterogeneously and asymmetrically distributed throughout the cell and constitutes more than 30% of the lipid on the inner leaflet of the plasma membrane; this, combined with its negatively charged head group, confers onto phosphatidylserine a nonpareil ability to direct the recruitment of proteins containing polycationic stretches as well as proteins that possess a specific PS-recognition site [86]. Under normal conditions, in the absence of environmental stress, negatively charged phospholipids are confined to the inner leaflet of the plasma membrane. However, changes in lipid asymmetry can expose these negatively charged phospholipids on the cell surface. The putative sensors of the Rim101 signaling pathway may identify the exposed negative charge similarly to how they identify the charge in alkaline culture medium [73]. Young et al. (2010) [87] demonstrated the role of phospholipids in pH signaling, specifically the function of phosphatidic acid as a cellular pH biosensor. The binding of proteins to the ubiquitous signaling lipid phosphatidic acid was dependent on the intracellular pH and protonation state of the phosphate head group. This evidence demonstrates that membrane asymmetry plays a crucial role in cell signaling and intracellular pH maintenance.

We have previously discussed the significance of membrane asymmetry as a signaling mechanism and adaptation to alkaline pH. However, recent reports indicate that membrane composition may also play a significant role in alkaline pH resistance. For instance, Danilova et al. (2020) [88] evaluated the composition of the cytoplasmic membrane of the haloalkali-tolerant fungus *Emericellopsis alkalina* at different pH (4.5, 7, 10.2), and they found that changes of the pH had little effect on the composition of membrane lipids. Only an increase in the proportion of phosphatidic acids in neutral conditions (pH 7.0) and an increase in the proportion of sterols in alkaline conditions (pH 10.2) were observed. Phosphatidic acid also increases when growing *Sodiomyces magadii*, *S. alkalinus*, and *S. tronii* at neutral pH, but according to the graphs shown by the authors, the sterols had a different behavior compared to *E. alkaline*; at alkaline pH, there is a decrease in the proportion of sterols [89,90].

This behavior of little fluctuation in membrane composition in response to pH and variation in phosphatidic acids and sterols was also demonstrated in the alkali-tolerant fungi *Acrostalagmus luteoalbus* and *Chordomyces antarcticus* [90]. The behavior of phosphatidic acid at different pH is similar between alkaliphilic and alkali-tolerant fungi, with a higher proportion in neutral conditions but a decrease in sterols at alkaline pH. The differences in sterols between the alkaliphilic/alkali-tolerant fungi (*S. magadii*, *S. alkalinus*, *S. tronii, A. luteoalbus*, and *C. antarcticus*) with *E. alkalina* may be because the latter was cultivated under saline conditions (0.4 M NaCl).

The total lipids in the membranes of alkaliphilic and alkali-tolerant fungi is another interesting feature to examine. Alkaliphiles exhibit an increase in total lipids at acidic/neutral pH, whereas alkali-tolerant organisms exhibit an increase at alkaline pH [90]. This is likely because alkaliphilic fungi experience stress in an acidic pH environment, whereas alkali-tolerant fungi thrive in an acidic pH environment. Although there is evidence of little change in the composition of the cytoplasmic membrane in response to pH, these changes in total lipids may indicate that we do not completely comprehend the membrane’s behavior at different pH levels.

### 3.3. Protein Adaptations

During evolution, organisms achieved viability under extreme conditions either by ‘escaping’ or ‘compensating’ the stress or by enhancing the stability of their cellular inventory [91]. Although a great deal is known about the adaptation of proteins to elevated temperatures [92,93] and salinity [94,95], there are few comparable studies on the effect of pH in fungi. In addition, the majority of the prior research on alkaline pH stability has been conducted on bacterial proteins [10], whereas reports on fungal proteins are scarce and based on studies of enzymatic activities in mesophilic fungi. We are unaware of any enzymatic investigations on alkaliphilic or alkali-tolerant fungi that explain their activity at alkaline pH. It would be premature to discuss the causes within fungi that determine the stability of acidophilic and alkaliphilic proteins. However, research indicates how these potential determinants may be.

The pH activity profiles of enzymes are highly dependent on the pKa of the catalytic residues, which are themselves dependent on the local environment and, thus, on the nature of the amino acids in the vicinity of the catalytic residues [96]. Furthermore, some data indicate a role for hydrogen bonds and salts bridges in protein stabilization [97,98].

It is likely that some amino acids have an important role in catalytic activity and in enzyme stability in alkaline pH. Wang et al. (2005) [99] analyzed the pH profile of endoglucanase III from neutrophilic *Trichoderma reesei* to determine the effect of substitutions at surface residue N321, where two catalytic residues are located. N321D mutation caused a significant shift in the optimum pH of the enzyme from 4.8 to 4.0, whereas N321H maintained its highest activity in a pH range from 4.6 to 5.5 and more than 84% of its specific activity at pH 6.0. Positive charges near the catalytic cleft probably promote enzyme activity at higher pH, but there is currently no evidence to support this hypothesis in enzymes isolated from alkaliphilic or alkali-tolerant fungi.

Other examples are certain xylanases that are stable in alkaline conditions, which are typically characterized by a decreased number of acidic residues and an increased number of arginine [100]. Substitutions of Ser/Thr surface by arginine residues in *Trichoderma reesei* endo-1,4-β-xylanase II shifted the activity profile to the alkaline region by 0.5–1.0 pH units [100]. Luo et al. (2012) [101] studied the alkali-tolerant β-mannanases of *Humicola insolens* Y1, which displays high activity at pH 9.0 and over 10% at pH 10.0, in order to determine the particularities that allowed it to resist alkaline pH. The β-mannanases of *H. insolens* Y1 have 11% alkaline amino acid residues (Arg and Lys), compared to the acid-active mannanases (3.7–5.3%), and most of the alkaline residues are located on the protein surface. It is probable that the higher pKa values, high frequency of alkaline amino acid residues, and more positively charged residues might be the factors that influence the protein activity and stability under neutral and alkaline conditions [101].

The formation of hydrogen bonds between amide–carboxylate pairs can play an important role in the stability of proteins. Substitutions of carboxyl–carboxylate pairs to amide–carboxylate pairs in *Trichoderma reesei* cellobiohydrolase showed an increase in protein stability at alkaline pH. This is probably because the carboxyl–carboxylate pairs are negatively charged at alkaline pH, and the strong repulsion causes conformational changes that weaken the stability of the protein. In contrast, amide–carboxylate pairs can maintain the hydrogen bond at different pH [102].

According to the evidence, the primary strategies for resisting alkaline pH at the protein level are variations in pKa, the formation of hydrogen bonds between amide–carboxylate pairs, and the presence of alkaline amino acids on the surface of proteins. However, amino acid substitution experiments shifted the optimum enzyme pH by less than one unit of pH towards alkalinity. In addition to the interactions discussed in this section, it is probable that other interactions contribute to the stability required to withstand high pH values.

## 4. Biotechnological Impact of Alkaliphilic and Alkali-Tolerant Fungi

Extremophiles, uncommon microorganisms that thrive in extreme environments of temperature, pH, salt, sugar concentrations, and atmospheric pressure, have been identified over the past four decades [10]. Extremophile microorganisms provide robust enzymatic and whole-cell biocatalytic systems that are alluring under conditions that limit the efficacy of conventional bioconversions [4]. The extraordinary survival strategies of extremophiles can be linked to the production of extremozymes and other secondary metabolites with numerous biotechnological applications [103,104]. Fungi must be able to generate and secrete a large number of hydrolytic enzymes that degrade plant organic matter (e.g., maize, wheat, or rice straw), such as cellulose, hemicellulose, and lignin, in order to proliferate in soil [34]. In the case of alkaliphilic and alkali-tolerant fungi, they are known to produce alkaline proteases, cellulases, chitinases (Table 1), and other biotechnologically significant metabolic products, such as carotenoids, steroids, antibiotics, and organic acids [9]. Alkali-resistant enzymes isolated from mesophilic fungi also have biotechnological potential, constituting a second relevant group.

In general, microorganisms must exist in environments where the nutrients are primarily macromolecular in nature. These nutrients are inaccessible to microbes unless they are broken down into molecules that they can assimilate. An enzyme secreted by microorganisms breaks down macromolecular nutrients into smaller molecules. Alkaline proteases have proven beneficial in the detergent, food, pharmaceutical, and tanning industries [105].

The most practiced method for the depilation of hides and skins is the lime–sulfide process; because sulfur is toxic, enzymatic depilation has gained more acceptance [106]. The alkaline protease from *Aspergillus flavus* was obtained by solid-substrate fermentation and its characteristics allow it to be used for this purpose [107]. A similar protease was isolated from *Aspergillus clavatus*; its high proteolytic activity at alkaline pH, at moderate temperature and stability towards various surfactants and bleaching agents, suggest its suitability for inclusion in liquid detergent compositions [108]. Detergents are chemical compounds used for cleaning purposes. However, stains of blood and soil being proteinaceous in nature are not easily removed by detergent chemicals. Therefore, proteases have an important role in enhancing detergent cleaning performance [10]. Therefore, identifying, isolating, and characterizing alkaline proteases which do not lose their activity in detergents is of interest to biotechnologists.

**Table 1 jof-09-00652-t001:** Review of alkaline enzymes and alkali-resistant enzymes found in the literature.

Enzyme	Microorganism	pH Enzyme Production	Optimum pH for Activity	Optimum T (°C) for Activity	References
Alkaline protease	*Aspergillus flavus*	7	Optimum activity in the pH range of 7.5–9.5.	Optimum activity at 42 °C.	[107]
	*Neosartorya clavata* (syn. *Aspergillus clavatus*)	8	Optimum activity at pH 8.5. It is active in the pH range of 6.0–11.0. The enzyme is stable after an hour of incubation at pH 8.0–9.0.	The enzyme was active between 40 and 70 °C with an optimum around 50 °C. Stable activity after 1 h treatment at 30 °C.	[108,109]
	*Sodiomyces alkalinus*	10	Optimum activity at pH 8.	Optimum activity at 30 °C.	[64]
	*Emericella usta* (syn. *Aspergillus ustus*)	9	Enzyme active in a wide pH range (6.0–10.0), with an optimum at pH 9.0.	Optimum activity at 45 °C.	[110]
Serine protease	*Aspergillus parasiticus*	-	Optimum activity at pH 8. Stable activity after 1 h treatment in pH 6.0–10.0.	Optimum activity at 40 °C. The enzyme was stable at 40 °C for 1 h incubation but was inactivated at temperatures over 40 °C	[111]
	*Parengyodontium album* (syn. *Engyodontium album)*	10	Enzyme active in a wide pH range (6.0–12.0), with an optimum at pH 11.0.	Optimum activity at 60 °C. More than 90% of the maximal activity was conserved between 45 and 65 °C.	[112]
	*Clonostachys rosea* (syn. *Gliocladium roseum*)	7–7.5	Optimum activity in the pH range of 9.0–10.0.	Optimum activity at 60 °C.	[113]
	*Trichoderma reesei*	6	Enzyme active in a wide pH range (6.0–11.0) with an optimum at pH 8.0.	Optimum activity at 50 °C.	[114]
Dipeptidyl peptidase 4	*Chordomyces antarcticus*	10	Optimum activity at pH 7.7. Stable in the pH range of 3.0–12.0.	Optimum activity at 37 °C.	[115]
	*Sodiomyces alkalinus*	10	Optimum activity at pH 7.3. Stable in the pH range of 5.0–12.0 and retained 23% of its activity after an hour of incubation at pH 13.0	Optimum activity at 37 °C.	[115]
Trypsin-like protease	*Cordyceps militaris*	6	Optimum activity in the pH range of 8.5–12.0.	Optimum activity at 25 °C.	[116]
Xylanase	*Penicillium citrinum*	-	Optimum activity at pH 8.5.	Optimum activity at 50 °C.	[117]
	*Cladosporium oxysporum*	7–8	Optimum activity at pH 8.0. Stable activity after 2 h treatment in pH 7.0–8.5.	Optimum activity at 50 °C. Stable activity after 2 h treatment below 55 °C.	[118]
	*Aspergillus fischeri*	9	Optimum activity pH 6.0. Retained over 50% of maximum activity at pH 8.0. pH stability ranged from 5.5 to 9.5 with retention of more than 85% of the activity.	The optimum temperature was 60°C.	[119]
	*Aspergillus fumigatus*	8	Optimum pH 8. Substantial residual activity at alkaline pH 8–9 (56–88%).	Optimum temperature 50 °C. Substantial residual activity at 60–70 °C (53–75%).	[120]
Amylases	*Clavispora lusitaniae*	7	Optimum activity at pH 11. Enzymes retained nearly 80% of activity after being exposed to various detergent components for 2 h.	Optimum activity at 40 °C. The enzyme retained 45% and 98% of their maximum activity at 4 °C and 25 °C.	[121]
	*Sporormiella minima* (syn. *Preussia mínima*)		Optimum pH 9.	Optimum activity at 25 °C.	[122]
Endoglucanase B	*Aspergillus niger * (The gene encoding the enzyme was cloned and expressed in *Pichia pastoris*)	-	Stable for 2 h at alkaline pH (7–10).	Stable for 3 h at temperatures below 60 °C.	[123]
Endoglucanases	*Mycothermus thermophilus* (syn. *Humicola insolens*) (The gene encoding the enzyme was cloned and expressed in *Aspergillus oryzae)*	-	Optimal activity between pH 7 and 8.5.		[124]
Cellobiohydrolase II		-	Optimal activity pH 9.		
Laccase	*Albifimbria verrucaria* (syn. *Myrothecium verrucaria*)	9	Optimum activity at pH 9.0 and retained more than 80% of the initial activity after 17 h incubation at 30 °C at pH 8–11.5.	Stable for 1 h at temperatures below 50 °C and retained more than 80% of the initial activity.	[125]
Lipases	*Trichoderma lentiforme * (The gene encoding the enzyme was cloned and expressed in *Pichia pastoris*)	-	Poor activity under acidic conditions and maximum activity at pH 9.5. The enzyme was stable at pH 6.0–9.0, retaining more than 80% of the initial activity after 1 h pre- incubation at 37 °C.	Temperature optimum at 50 °C and the enzyme was relatively stable at 40 °C, retaining more than 60% of the initial activity after 1 h incubation.	[126]
	*Aspergillus carneus*	8	The purified enzyme could tolerate pH 6.0–12.0 and it was stable in this range for 24 h. The optimum pH was 9.0.	The optimum temperature was 37 °C and the enzyme was active in the range of 5- 90 °C.	[127]
	*Lasiodiplodia theobromae*	8	Optimum activity pH 8.0.	Optimum temperature was 30 °C.	[128]
	*Gliocladium sp.*	-	Optimum activity pH 10.	The assay temperature was 40 °C.	[129]
	*Verticillium sp.*	-	The enzyme was fully stable for 30 min at pH 9 at temperatures up to 50 °C. The enzyme was stable throughout the pH range 6–10 at 25 °C for 24 h.		[129]

The discovery of alkaline cellulases allowed for the formulation of cellulase-containing detergents. There are many bacterial cellulases that, unlike other detergent enzymes, do not degrade or modify the dirt or stains; instead, these enzymes modify the surface of the garment cellulosic fiber to improve the overall washing performance [130]. Filamentous fungi are excellent organisms that can function as cell factories for the production of a variety of products. They are robust and naturally produce efficient enzymes for the decomposition and conversion of biological material [131]; however, most fungal cellulases are active in acidic or neutral conditions [130], hence the importance of isolating and characterizing alkaline cellulases.

Another interesting application of cellulases is in the textile industry. These enzymes have been incorporated into the refining process of cellulose-based textiles in an effort to enhance the feel and appearance of the textiles [132]. Although neutral and acid active cellulases are used in the biostoning process, alkaline active cellulases have been suggested for this process because they prevent backstaining [133]. Endoglucanase B identified in *Aspergillus niger* was cloned and expressed in *Pichia pastoris* [123]; the enzyme activity was increased by 2-fold by Triton X-100 and Tween 80. Endoglucanase B is a potential candidate for use in laundry and textile industrial applications due to its favorable properties. The paper recycling process includes alkaline pH and temperatures close to 60 °C [134]; therefore, the use of alkaline and thermostable cellulases could enhance toner removal from wastepaper and facilitate the recycling process.

Similarly, laccases can also be used in the paper industry to decolorize and decompose dyes and phenolic compounds or to remove toxic aromatic compounds in wastewater from various industries [125]. Therefore, laccase from *Albifimbria verrucaria* (syn. *Myrothecium verrucaria*) can be used in those scenarios (Table 1).

On other hand, recently, alkaline xylanases have attracted increasing interest in their applications in paper and feed industries, textile processes, enzymatic saccharification, and waste treatments [118]. The modern pulp and paper industry employs various techniques for recovering cellulose fibers from lignocellulosic biomass, the majority of which is derived from wood. This procedure employs alkaline and high-temperature treatments to degrade and solubilize lignin, which must then be eradicated via a chlorine-based bleaching procedure [130]. The use of chlorine results in the formation of highly toxic and mutagenic chlorinated organic by-products [135]. Xylan hydrolysis facilitates lignin removal; therefore, enzymatic treatment permits a substantial reduction in chlorine use [136]. In addition, alkaline xylanase could treat material with a high pH without the need for costly and time-consuming pH adjustments [118]. The xylanases listed in Table 1 have potential industrial applications in textile and paper manufacturing.

Excessive salt accumulation in soil imposes many negative effects on its organic matter decomposition and uptake of available nutrients [137,138]. Some halo- and/or alkaliphilic fungi are promising agents for bioremediation of saline and alkaline soils. In addition, these fungi can be considered as genetic pools that represent a resource for cloning genes related to saline/alkali-resistance or tolerance [139] and organic matter degradation, which can be used to improve or create highly active fungi for soil remediation [34]. Another attractive property of the alkaliphiles is their ability to produce organic acids. *Aspergillus glaucus* is an haloalkaliphilic fungus that produces a variety of organic acids, including citric acid, oxalic acid, and malic acid [140]; therefore, they can be used for soil mycoremediation [34]. *A. glaucus* was evaluated for the bioremediation of saline–alkaline soil in the Songnen plain of Northeastern China. This study primarily indicates that the applied amendments mixed with haloalkaliphilic fungi significantly encourage steady growth and yield of rice in comparison with that achieved in the control plot [141].

Another group of less-studied fungal alkaline enzymes and of great importance are starch degrading enzymes [130]. Amylases have applications in starch processing [142], bakery and sugar production [143], desizing in textile industries [144], and detergent manufacturing [145]. The biochemical characteristics of amylases from *Clavispora lusitaniae* and *Sporormiella minima* (Table 1) suggest that theses enzymes may find application in the laundry detergent and textile industries [121,122]. The fungal amylases are preferred over other microbial sources because fungi are generally safer, by the hyphal mode of growth, and their good tolerance to low water activity and high osmotic pressure conditions make fungi most efficient for bioconversion of solid substrates [146].

Lipases are a group of enzymes with significant physiological and industrial significance [147]. Lipases catalyze the hydrolysis of triacylglycerols to glycerol and free fatty acids, and in contrast to esterases, they are only active when adsorbed to an oil–water interface [148]. Lipases are prevalent in nature and have been identified in microorganisms, plants, and animals [149,150,151]. Nonetheless, fungal lipases are of particular interest because they are readily produced and advantageous for industrial applications due to their high yields, versatility, and resistance to harsh conditions. Lipases have promising uses in organic chemical processing, detergent formulations, biosurfactant synthesis, the oleochemical industry, the dairy and agrochemical industry, paper manufacturing, nutrition, cosmetics, and pharmaceutical processing [147]. Most of these conditions necessitate detergent-, low-temperature-, and alkaline-pH-resistant lipases; therefore, their isolation and study are necessary.

The properties of alkaline lipases shown in Table 1 allow them to be used in the washing industry. However, the interest in employing lipases to catalyze reverse synthesis reactions (esterification, transesterification) in non-aqueous environments has increased markedly due to their application in the production of biodiesel [152]. Venkatesagowda et al. (2017) [128] evaluated the potential of *Lasiodiplodia theobromae* lipases, demonstrating their potential for biodiesel production from coconut oil in alkaline conditions.

The screening of extremophiles may also yield other bioactive compounds that can be used in the treatment of non-infectious diseases, including cancer and cardiovascular problems [130]. Rogozhin et al. (2018) [153] studied the ability of the alkaliphilic fungus *Emericellopsis alkalina* VKPM F1428 to produce antimicrobials. They were able to isolate and identify a Lipopeptaibol Emericellipsin A with antimicrobial activity against fungi and Gram-positive bacteria. They also demonstrated in vitro selective cytotoxic activity against HepG2 and Hela cell lines. Subsequent work on this strain showed that it produces four additional homologues (Emericellipsin B-E) and that Emericellipsin A has activity against clinical isolates of *Aspergillus* and yeasts [154].

Kuvarina et al. (2021) [155] studied the antimicrobial-producing potential of 25 alkaliphilic fungal strains belonging to *Sodiomyces alkalinus*. Some of these strains showed antimicrobial activity against the opportunistic pathogens *Aspergillus niger* INA 00760, *Bacillus subtilis* ATCC 6633, and *Candida albicans* ATCC 2091. In one of these strains, *S. alkalinus* 8KS17-10, a class II hydrophobin with antifungal activity against non- and clinical pathogenic isolates was purified and characterized [156]. We submitted the *S. alkalinus* genome (BioSample: SAMN02745840) to Antismash [157] to predict biosynthetic gene clusters and assess their potential as a producer of bioactives. The results showed 24 biosynthetic clusters that may be related to Polyketide synthases, Nonribosomal peptide synthetase, ribosomally synthesized and post-translationally modified peptides, and terpenes (Appendix A). Many of these clusters are likely to be novel and their products could have higher activity than current bioactives.

Extremely diverse enzymes are produced by alkaliphilic fungi, although the production and commercialization of these enzymes are currently centered on bacteria. The scientific community remains attracted to the use of fungi for the production of alkaline and alkali-resistant enzymes and their subsequent industrial application, particularly since many of these enzymes are extracellular, which facilitates their purification by filtration. This stimulates the search for alkaliphilic fungi, the investigation of their survival mechanisms, and the identification of pH-resistant exoenzymes.

## 5. *Aspergillus sydowii*, a Case Study

We have discussed the significance of alkaliphilic and alkali-tolerant fungi in biotechnology throughout this article. However, it is fascinating to provide a further illustration. *A. sydowii*, a halophilic fungus capable of expressing enzymes with biotechnological potential, was isolated in our laboratory under salinity conditions [158]. It produced cellulases, xylanases, manganese peroxidase, and esterase. González-Abradelo et al. (2019) [159] evaluated the growth of *A. sydowii* for the removal of polycyclic aromatic hydrocarbons (PAHs) and pharmaceutical compounds. This strain used benzo-α-pyrene and phenanthrene as the sole carbon source in a mineral medium, biodegrading both PAHs (99% benzo-α-pyrene and 97% phenanthrene). In addition, *A. sydowii* removed (biodegradation and adsorption) more than 90% of the pharmaceutical compounds evaluated (acetaminophen, mefenamic acid, ketoprofen, indomethacin, and ibuprofen). Alkaline and salty environments could be interrelated [5], hence the significance of determining this strain’s alkaline pH resistance.

*A. sydowii*’s growth on Mineral Medium Agar [160] with 2% glucose (*w*/*v*) was evaluated. The medium was supplemented with various NaCl concentrations (0–1.5 M) and pH (4–11.2). To maintain the pH, buffers with a final concentration of 0.2 mM were added to the mineral medium. The buffer choices for generating different final pH values were Na_3_C_6_H_5_O_7_/citric acid buffer system for pH 4 and 5, Na_2_HPO_4_/NaH_2_PO_4_ buffer system for pH 6.2, 7, and 8, Na_2_CO_3_/NaHCO_3_ carbonate buffer system for pH 9.2 and 10, and Na_2_HPO_4_/NaOH buffer system for pH 11.2 [5]. To ascertain the growth rate (mm/day), we measured mycelium growth daily. The results show that *A. sydowii* has a better growth at acid–neutral pH; however, it can resist alkaline pH values, evidenced by its growth up to pH 11.2 (Figure 5). Compared to pH 11.2, growth is significantly inhibited at pH 9.2 and 10; however, we believe that this inhibition is not solely attributable to pH. There are reports of the use of sodium carbonate as an antimicrobial [161], and the Na_2_CO_3_/NaHCO_3_ carbonate system was used to evaluate growth at pH 9.2 and 10. We conclude that *A. sydowii* is alkali-tolerant under our experimental conditions, particularly if we analyze the growth at 0 M NaCl condition, in which there are no statistically significant differences between pH 7, 8, and 11.2. Adding NaCl to the culture medium decreases the growth rate considerably at pH 11.2, indicating that the combined effect of both stresses (salinity and pH) inhibits the growth of *A. sydowii*.

Previous evidence shows that *A. sydowii* is halophilic [162]; however, according to our data, the behavior of *A. sydowii* is halotolerant. This discrepancy is due to differences in cultivation methods. Jiménez-Gómez et al. (2022) [162] used a rich medium (Yeast Malt Agar) supplemented with 1 M NaCl, and in our growth kinetics, the medium used was Mineral Medium Agar supplemented with glucose and NaCl. This difference causes changes in the specific growth rate of *A. sydowii*, especially since in the mineral medium the stress conditions are stronger compared to the Yeast Malt Agar.

Additional studies should be carried out on *A. sydowii* to determine which other xenobiotics it can degrade or to understand the mechanisms that allow it to resist alkaline pH. However, current studies reaffirm that the isolation and characterization of halophilic and alkali-tolerant fungi such as *A. sydowii*, with the ability to degrade xenobiotics under extreme conditions, opens doors to the development of new methodologies for bioremediation. *A. sydowii* could be an excellent candidate to study the adaptations involved in both extreme growth conditions, NaCl and alkaline pH.

## 6. Conclusions

Bacteria have been the primary focus of research into organisms that are resistant to environmental pH changes. Recent research reveals that a vast array of fungi can resist both alkalinity and salinity. In recent years, there has been an increase in interest in these organisms as a source of alkaline enzymes for biotechnological applications. Although alkaline pH signaling mechanisms in alkali-tolerant fungi have been investigated, it is still unclear how these fungi maintain intracellular homeostasis and what properties enable their extracellular enzymes to remain active at alkaline pH.

## Figures and Tables

**Figure 1 jof-09-00652-f001:**
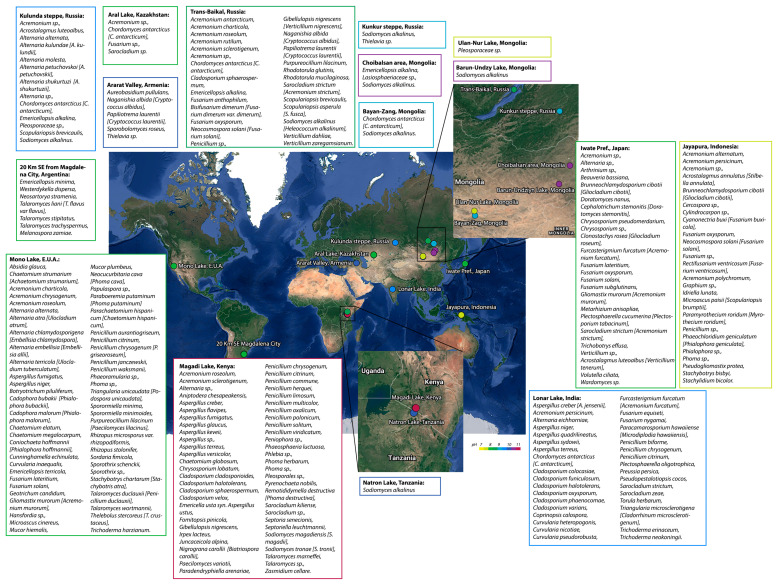
Distribution of alkaline environments around the world (colored circles). Fungi species isolated from alkaline ecosystems are shown. Species nomenclature was updated according to the Mycobank Database. The species name given in the cited article is enclosed in brackets. The color of the circles and the frame that encloses the names of the species indicate the average pH of the isolation sites. The following references were used for the figure: Kulunda steppe [5,18,19]; Aral Lake [5,19]; Ararat Valley [5,20]; Trans-Baikal [5,19,20,21]; Kunkur steppe, Choibalsan area, and Bayan-Zang [5,18]; Ulan-Nur Lake [5]; Barun-Undzy Lake and Natron Lake [18]; Iwate [22]; Jayapura [23]; Lonar Lake [24]; Magadi Lake [5,25,26]; Mono Lake [17]; 20 Km SE from Magdalena City [27]. Studies [5,18,19,24,25,26] used morphophysiological and molecular genetic characterization as taxonomic criteria, while studies [17,20,21,22,23,27] used only morphophysiological characterization for fungal identification.

**Figure 2 jof-09-00652-f002:**
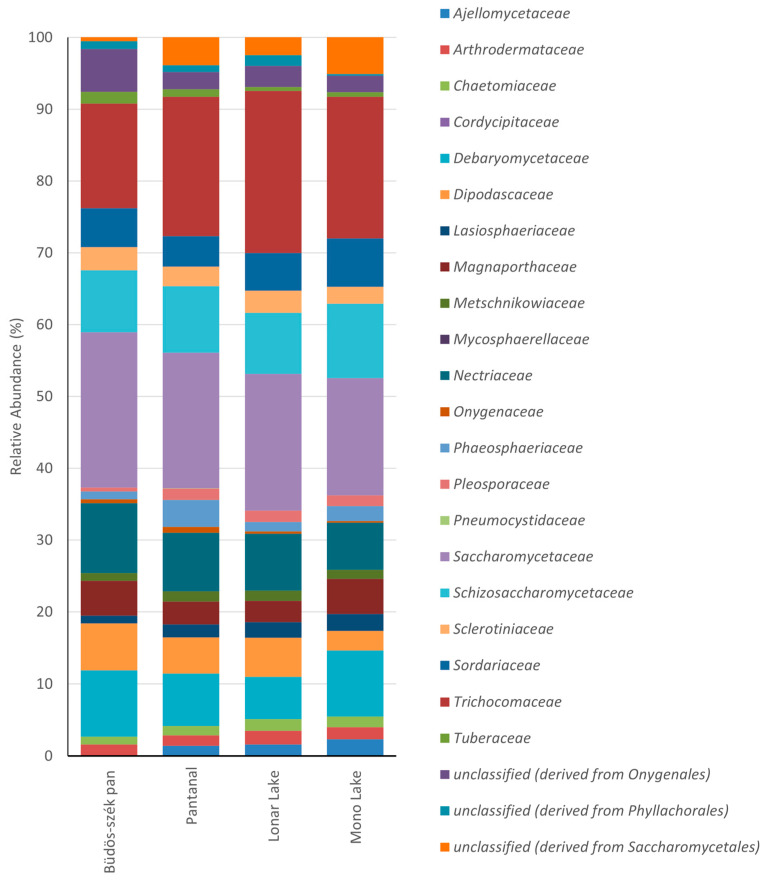
Relative Family abundance of *Ascomycota* for the four metagenomes: Büdös-szék pan (MG-RAST ID: mgm4555202.3), Pantanal (MG-RAST ID: mgm4875736.3), Lonar Lake (MG-RAST ID: mgm4863970.3), and Mono Lake (MG-RAST ID: mgm4962892.3). Abundance analysis was performed in MG-RAST (www.mg-rast.org (accessed on 7 May 2023)) with the default values that are in the program. Abundance values are in the Appendix A.

**Figure 3 jof-09-00652-f003:**
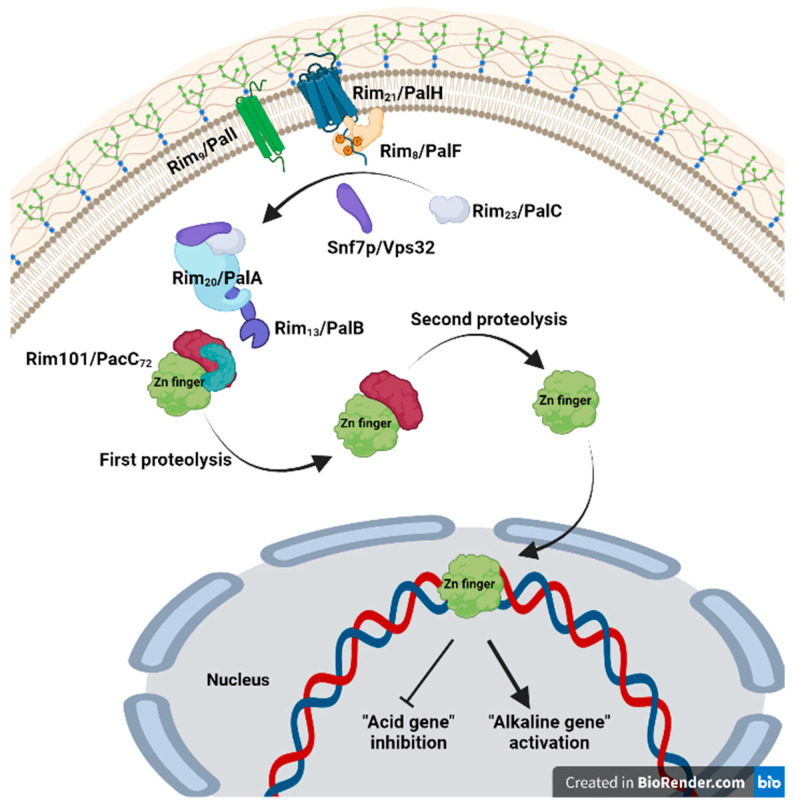
Signal transduction pathway for the activation of the Rim101/PacC transcription factor. The figure was created in Biorender.com (accessed on 7 May 2023).

**Figure 4 jof-09-00652-f004:**
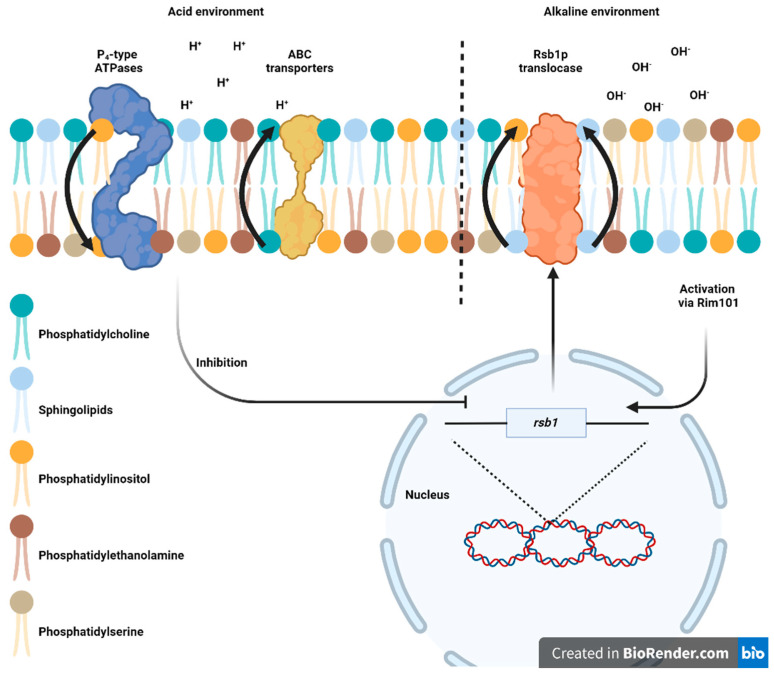
Effect of pH on the asymmetry of the cytoplasmic membrane of *Saccharomyces cerevisiae* and signaling mechanisms to conserve lipid distribution in the membrane. The image was designed taking into account the evidence obtained in references [73,84]. The figure was created in Biorender.com (accessed on 7 May 2023).

**Figure 5 jof-09-00652-f005:**
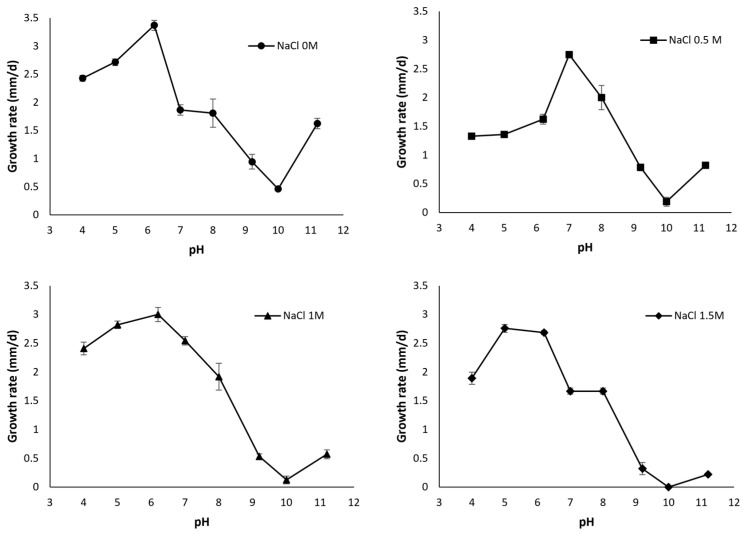
Effect of pH on the growth rate (mm/d) of *Aspergillus sydowii* in mineral medium with glucose (2%) and different concentrations of NaCl.

## Data Availability

The data presented in this study are available in both the article and in the Appendix A.

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
