# Peer review of "Alkaliphilic/Alkali-Tolerant Fungi: Molecular, Biochemical, and Biotechnological Aspects"

_jof, 2023, doi:10.3390/jof9060652_

Round 1
Reviewer 1 Report
The subject of the manuscript is interesting and relevant, since there is a need to systematize the accumulated experimental data in this area.
The stated chapters correspond well to the topic of the review, but the content of these chapters itself requires significant revision.
Introduction
line 30: Macelroy in 1974 – bibliography number is not specified (it should be “2”).
lines 35-37, reference to 6 is not appropriate in this context, since the review of Mamo @ Mattiasson is primarily about prokaryotes, not fungi; the criterion given in that work reflects the division of groups of prokaryotes according to the type of adaptation to pH.
Since the manuscript is devoted to alkaliphilic fungi, it is necessary to focus on fungi.
Accordingly, it is necessary to describe in the introduction the criteria for dividing fungi into alkaliphilic and alkalitolerant ones. This should be clearly stated in order to understand which group of fungi is the subject of this manuscript. Since this is a review, it is recommended to compare the criteria for dividing into groups according to pH adaptation by different authors. Next, it is necessary to formulate and justify the definition of alkaliphilic fungi, which will be used in this manuscript.
There are contradictory statements in the manuscript:
line 11: «alkaliphilic and alkali tolerant fungi, which resist pH above 9»
lines 35-36 – «The group of microbes that grow optimally below pH 9 but thrives well in high pH environment (> pH 9) are known as alkali-tolerant»
line 79 «Most alkaliphiles have an optimal growth pH of around 10» etc.
Therefore, in the introduction, the criteria for alkaliphilic fungi must be formulated, and further in the text, it is necessary to adhere to them.
line 38 reference to 7. This work concerns mainly prokaryotes, what needs to be specified so that the reader does not have the mistaken impression that so much is also known about fungi.
line 44 – extra “;”
2. Alkaliphilic fungi: distribution and isolation
The section is very important within the framework of the stated topic, since the manuscript should really provide information on the distribution and diversity of alkaliphilic fungi.
However, the authors failed to clarify the subject matter.
There is much more research on fungi both in soda soils/lakes and in anthropogenic biotopes. But after reading the section, there is a feeling that there are only two articles - No. 5 (including No. 15, since this is the work of the same group of researchers) and No. 17.
It is not very clear why the authors of the manuscript classify the fungi isolated from the coast of Lake Mono as alkaliphilic. Firstly, the isolation medium was not prepared on a buffer basis, and it is known that fungi can significantly change the pH of the medium, and therefore, when studying the relationship to the pH, it is very important to use buffer media. Secondly, in the Mono Lake fungi study, there was no study of the pH adaptations of pure cultures. While simply isolation on an alkaline medium (in this case, not even on a buffer basis) is not a reason to classify the strain as alkaliphilic. Not all fungi, even among those isolated on alkaline buffer media, turn out to be alkaliphilic upon detailed study of pure cultures. Most fungi in such biotopes are alkalitolerant.
And also, in order to claim that the fungus is an alkaliphile, in addition to a thorough study of pH adaptation, it is also necessary to be confident in the identification of a culture. And for this it is necessary that it be determined not only by morphological and cultural characteristics, but also by at least one genetic locus.
As for the reference No. 5, Table 1 contains fungi that the authors of the cited article do not classify as such. Besides, a separate selection in the table of Sodiomyces alkalinus with a citation of No. 15 is redundant, since the same isolates were included in No. 5. That is, in Table 1 these species are given twice, first in a short version, with reference to No. 15, and then in a more detailed version, with reference to No. 5.
Table No. 1 cannot be published as presented, because it is 4.5 pages long, its content does not match the title, and the table is based on the analysis of only two articles. The column "Growth conditions" reflects the conditions for isolating pure cultures, and not their possible growth conditions. Also, the names of specific geographical places take up a lot of space in the table, and further in the text of the manuscript is not discussed at all. Giving such detailed geographic names seems redundant within the framework of this manuscript.
The end of the section describes the alkaline conditions that can be formed by human activities, and the section concludes with a general discussion of the potential of organisms living in alkaline conditions. It is necessary to add references to works devoted specifically to alkaliphilic fungi isolated from such places. What species are most often isolated? A comparison should be made of communities of alkaliphilic fungi isolated from natural and anthropogenic habitats.
It's a good idea to represent soda salt flats and the variety of fungi in them on the map (Fig.1 «Worldwide distribution of some soda lakes and deserts. The image shows the fungi isolated from these sites and the colors show the pH of the isolation site»). But to make it clear why there are deserts in the title of the picture, it would be necessary to explain earlier that in this zone there is often soda salinization.
It is not clear from what sources and according to what criterion the species shown in the boxes in the figure were selected. It seems that these are not strictly alkaliphiles, but perhaps alkalitolerants or just fungi from these biotopes? The given lists of fungi do not correspond to the articles previously cited in this section, and they are not shown earlier in Table 1 either. For example, we see Cryptococcus, Rhodotorula (Rhodotorula mucilaginosa is misspelled), and others, although nothing has been written about yeasts before, and no references are given here. It is also necessary to correct outdated species names, for example, Heleococcum alkalinum is now Sodiomyces alkalinus, as indicated in No. 5 and 15.
When citing the species of fungi in a table 1 or in figure 1, it is necessary to check their modern taxonomic names. Those species whose identification is confirmed not only by morphological characters, but also by molecular-genetic ones, should also be noted separately.
Figure 1 needs significant improvement. But if the authors of the manuscript won’t offer some new version of Table 1, the corrected figure 1 for this section of the manuscript is sufficient.
Thus, if this section remains, then it should be carefully revised, and it is necessary to add to the discussion additional data on alkaliphilic fungi both from various natural biotopes and from anthropogenic conditions. For example:
Elíades, L. A., M. N. Cabello & C. E. Voget. 2006. Contribution to the study of alkalophilic and alkalitolerant Ascomycota from Argentina. Darwiniana 44(1): 64-73.; Sharma et al., 2016, https://doi.org/10.3389/fmicb.2016.01847
Orwa et al., 2020 https://doi.org/10.1016/j.heliyon.2019.e02823
Nagai K., Suzuki K., Okada G. Studies on the distribution of alkalophilic and alkali-tolerant soil fungi II: Fungal flora two limestone caves in Japan. Mycoscience. 1998. N. 39. P. 293–298.
Authors should also use data from other works, including those already cited in the list of references.
3. Physiology
In the first sentence, a new definition of alkaliphiles is given, without reference. Once again I emphasize the importance of defining the term "alkaliphilic fungi" in the introduction, specifying whether the authors of the manuscript accept the division into obligate and facultative alkaliphiles, as it is customary for prokaryotes and is used in some works on eukaryotes as well.
The authors of work 22 do not in any way associate the acids secreted by neutrophilic fungi of the rhizosphere with resistance to pH of the medium. It is necessary to discuss works on the ability of fungi to synthesize acids in alkaline biotopes, if there are such studies.
Reference 24 refers to prokaryotes, not fungi.
Accordingly, the introduction to Section 3 should be formulated more clearly, and it is necessary to separate what is known for alkaliphilic prokaryotes and neutrophilic fungi, what is known for alkaliphilic fungi, and what is only suggested as possible mechanisms of resistance to high environmental pH in fungi.
3.1. Genes, transcription factors and signaling mechanism
As the authors rightly pointed out, the alkaliphiles differ significantly from the commonly studied neutrophils (lines 79-80). However, in section 3.1, devoted to the characterization of such important adaptive mechanisms as the Pal/Rim pathway, only data on two well-known model objects that are not alkaliphiles are discussed. Where were these fungi originally isolated from? What type of pH adaptation do they have? To what extent is it possible to transfer the features of the Pal/Rim pathway in these fungi to other species? Most of the work discussed in this section refers to the early 2000s, what has been done on this topic since then? For which fungi have Pal/Rim pathways been studied? Are there works devoted specifically to obligate alkaliphiles? What was shown in comparison with traditional model objects?
line 132: Lucena-Agell et al. (2015) – bibliography number is not specified (it should be “44”)
line 155: Xu et al. (2004) – bibliography number is not specified (it should be “48”?)
Fig. 2 must be raised higher in the text (to the first link).
I believe that the section requires significant revision and data on studies of alkaliphilic fungi needs to be added (since the title of the manuscript is alkaliphilic fungi).
3.2. Membrane adaptations
line 224: Young et al. (2010) – bibliography number is not specified (it should be “68”)
The section is mainly devoted to general work on the cytoplasmic membrane, however, in the text of the manuscript, it is necessary to indicate species discussed. Judging by the references, the main part of the data concerns neutrophilic Saccharomyces cerevisiae. Then it is necessary to justify what relation these works have to the subject of the manuscript “Alkaliphilic fungi”. Only at the end of the chapter there is reference to the article (Young et al., 2010), which is close to the topic of the manuscript, but it also deals with regulation at acidic rather than alkaline pH.
It is also necessary to clarify which articles were the main ones for Fig. 3, which clearly shows the differences in membranes and signaling mechanisms under conditions of high and low pH. What species is this typical for?
It is necessary to revised section 3.2. in accordance with the subject of the manuscript and to add modern works on alkaliphilic fungi. For example, Bondarenko et al. 2018, Danilova 2020 and other works.
3.3. Protein adaptations
line 237 (Jaenicke 1998) – the article is absent in reference list.
Wang et al. (2005) – bibliography number is not specified (it should be “76”)
Indeed, there is not enough work on this topic.
Regarding the study by Wang et al. (2005), they worked with endoglucanase III from Trichoderma reesei under acidic conditions rather than alkaline. And, apparently, the fungus is also not an alkaliphile, then the discussion of it in this manuscript should be clarified.
Reference 77 concerns prokaryotes. The data obtained cannot be generalized to eukaryotes, and cannot be the result of a discussion of work on endoglucanase III from Trichoderma reesei. This can only be stated as a possible suggestion, with the proviso that data only exist for prokaryotes.
Also, regarding the other fungi discussed in this section, it is necessary to clarify where they were isolated from and what type of adaptation they have to the pH of the environment. If there are no studies on alkaliphilic fungi and that is why the text discusses the results of studies on neutrophilic species (Trichoderma reesei and Humicola insolens), then this should be written. If such studies exist, then, of course, they should be presented.
4. Biotechnological impact of alkaliphiles fungi
Table 2 requires correction and uniformity; the strain number should be given everywhere or nowhere.
Table 2 provides a good summary of the data on alkaliphilic enzymes, however, there is no explanation of how this relates to the topic of the manuscript "Alkaliphilic fungi". Are all the fungi listed in Table 2 alkaliphiles? If so, why are there no such species in section 2 of the manuscript? Alkalophilic enzymes are not equal to alkaliphilic fungi, and in relation to the subject of this manuscript, it is necessary to add an additional column indicating the pH adaptation of the described fungi.
The text of this section mainly contains general information about various enzymes of fungi, with a minimal touch on the alkali-resistant enzymes and enzymes of alkaliphilic fungi. Reviews on prokaryotes should be cited more carefully, without mixing with what is known about fungi. Discussion about the advantages of pH-resistant enzymes should be supported by references to studies on eukaryotes/ fungi. The text of the section can be correlated with the data given in Table 2, since alkali-resistant enzymes have already been found, and it would be correct to discuss them in the section on the biotechnological importance of alkaliphilic fungi.
line 338 «Another attractive property of the alkaliphiles is their ability to produce organic acids and change the medium pH [6], therefore they can be used for soil mycoremediation [23].» Review 6 refers to Exiguobacterium sp. as an example of such an organism and it is not correct to cite it here, since it is about mycoremediation. This may mislead readers about the degree of knowledge of alkaliphilic fungi in this area.
The following paragraphs about starch degrading enzymes (lines 340-345) and about Lipases (lines 346-357) also need to be supplemented with specific information on alkaliphilic fungi and alkaline enzymes, since so far only a general description of the areas of application of these enzymes has been presented, although Table 2 shows, for example, specific alkaline Lipases, and it would be appropriate to discuss them here.
Lines 343-344 «Also screening of extremophiles may also yield other bioactive compounds that can be used in treatment of non-infectious disease including cancer and cardiovascular prob lems [6]». The sentence is not correct in the paragraph about starch degrading enzymes. One could devote at least a separate paragraph to bioactive compounds obtained from alkaliphilic fungi, including antimicrobial peptides.
Thus, section 4 also should be revised. It is desirable to shorten the general discussion and citation of reviews on various groups of enzymes in fungi, and concentrate specifically on enzymes isolated from alkaliphiles and/or alkali-resistant enzymes, and discuss the possibility of their isolation from specific known alkaliphilic fungi. It would be interesting to compare the well-studied enzymes of alkaliphilic prokaryotes with modern data on these groups of enzymes in fungi (similarities, differences, areas of stability, areas of application, etc.).
5. A. sydowii, a case under study
lines 371-373 «Alkalophilic and saline environments are related [5], hence the importance of determining the capacity of this strain to resist alkaline environments». This sentence needs to be reformulated due to repetition «environments», and "environments" cannot be "alkalophilic" but only "alkaline".
line 389: González-Abradelo et al. (2019) bibliography number is not specified (it should be “116”)
References
No.15 – the reference is not correct.
No. 68 - the reference is not correct
As for the manuscript as a whole.
The text of the manuscript requires significant revision, with an emphasis on alkaline mushrooms and their specificity. At present, most of the text is devoted to neutrophilic organisms under high/low pH conditions or data on prokaryotes. The authors cite a large number of reviews, including those on prokaryotes, some of which can be shortened within the scope of the manuscript. It should be clearly indicated in the text on which organisms the study was conducted (prokaryotes or eukaryotes).
One of the known adaptive mechanisms of intracellular pH regulation is Na/H-ATPases, this mechanism is not discussed in the manuscript.
Throughout the text of the manuscript, uniformity in the use of terminology is required, because there are different spellings: "alkaliphilic" and "alkalophiles", as well as "alkali tolerant" and "alkalitolerant".
The proposed title «Alkaliphilic fungi: Physiology and biotechnological potential» is very broad, it may be better to change it to better reflect the actual content of the manuscript.
Author Response
Dear Reviewer 1,
Thank you for giving us the opportunity to revise this draft of the manuscript. We appreciate the time and effort that you have dedicated to providing your valuable feedback. We have been able to incorporate changes to reflect most of the suggestions provided by you and we have highlighted the changes within the manuscript.
We attach the file with the point-by-point responses to your suggestions.
Thanks in advance, kind regards.

Reviewer 2 Report
Alkaliphilic fungi are unique organisms with great potential for biotechnological use. The genome of these fungi can become a valuable source for further study of the evolution of the alkalophilic trait in fungi in relation to neutrophilic species. In addition, alkaliphilic fungi can provide the production of alkaline-active metabolites of commercial interest. From these positions, the topic of the review is interesting and useful.
Remarks:
Line 31 – add a reference to the article MacElroy, 1974
Line 58 - add an alkalotolerant to the title of Table 1, since many of the species considered in the table are alkalotolerants.
Lines 91-93 – note that these studies were conducted on neutrophilic species
Lines 177, 187, Fig. 3 – we are talking here about the P4 type of ATPases
Line 308 - peptidases should be added to Table 2, information about them is in the text, but for some reason is not in the table. Examples of articles where you can find data on such peptidases:
Grum-Grzhimaylo et al. The obligate alkalophilic soda-lake fungus Sodiomyces alkalinus has shifted to a protein diet. Molecular Ecology 2018;27: 4808–4819.
Alkin et al. Proline-Specific Fungal Peptidases: Genomic Analysis and Identification of Secreted DPP4 in Alkaliphilic and Alkalitolerant Fungi. J. Fungi 2021, 7(9), 744
Line 362 - fungal mycelium can be separated quite easily by simple filtration, but fungal proteins cannot be purified in this way
Line 388 - as follows from Fig. 4 A. sydowii halotolerant, not halophilic
Line 401 - one point on the graph (at pH 11.2) makes it difficult to judge the ability of A. sydowii to grow in a highly alkaline environment. Moreover, we see a steady decrease in the growth rate, starting from pH 7.0, not only in the carbonate buffer system, but also in the phosphate one. If the authors have doubts about the data obtained in the carbonate system, it is probably worth trying to replace this system with some other one.
Lines 26-27, 205-209 - redo, the authors' thought is not clearly expressed
Lines 155, 237, 249, 263, 389 – there are still references by authors, replace them with references in the order of citation and add them to the list of references
Author Response
Dear Reviewer 2,
Thank you for giving us the opportunity to revise this draft of the manuscript. We appreciate the time and effort that you have dedicated to providing your valuable feedback. We have been able to incorporate changes to reflect most of the suggestions provided by you and we have highlighted the changes within the manuscript.
We attach the file with the point-by-point responses to your suggestions.
Thanks in advance, kind regards.

Round 2
Reviewer 1 Report
The text has been significantly revised and improved.
However, some remarks remain.
1. The previous note is still relevant!
Point 31: Throughout the text of the manuscript, uniformity in the use of terminology is required, because there are different spellings: "alkaliphilic" and "alkalophiles", as well as "alkali tolerant" and "alkalitolerant".
Response 31: The text was homogenized with the terms alkalophilic and alkali-tolerant.
At the moment, the title of the article says "Alkalophilic / alkali-tolerant fungi", with "o". But almost EVERYWHERE in the text we see "alkaliphilic" with "i".
And also…
line 117 «alkaline-tolerant species»
line 472 – «alkalophilic enzymes»
line 572 «alkali tolerant» (without hyphen)
I strongly recommend checking the entire text again!!
2. As previously stated, "environments" cannot be "alkalophilic" but only "alkaline".
It is necessary to correct the text in line 13 "Alkaliphilic environments" and check throughout the text for similar typos.
3. I think it's also more correct to use "alkaline" and/or "alkali-resistant" for "enzymes" rather than "alkaliphilic".
At the moment, in Chapter 4 (and throughout the text of the article) both options are used, it is necessary to unify.
Here are a few of those typos:
line 457 «Alkaliphilic proteases»
line 469 «alkaliphilic proteases»
line 472 «alkalophilic enzymes»
4. After revision, section 2 looks much better, but it can still be improved.
Fig. 1 - in "Сhoibalsan area 5-18", there is a dash, not a comma.
The correct name for the fungus Chordomyces antarcticum is C. antarcticus.
Geat, that modern names of fungal species are added. Perhaps, in order to make it more clear where the old name is and where the modern one is, it is better to indicate the old name in brackets, while reducing the name of the genus to the first letter if it has not changed. In this case, in the caption to the figure, you can indicate that the name of the fungus in brackets is given as in the cited article.
There is at least one more work on alkali-tolerant fungi of Magadi Lake, moreover, and in this article the identification of fungi was carried out both by cultural and by molecular genetic features (Bondarenko et al, 2018; DOI: 10.1134/S1995425518050049).
5. Reference No. 15 from the first version of the manuscript is very important, but it was previously incorrectly listed in the list of references, one of the authors of the article was not indicated. I recommend to return the link to the section with the correct article output:
Grum-Grzhimaylo, A.A.; Debets, A.J.M.; Diepeningen, A.D. van; Georgieva, M.L.; Bilanenko, E.N. Sodiomyces alkalinus, a new holomorphic alkaliphilic ascomycete within the Plectosphaerellaceae. Persoonia 2013, 31, 147–158, doi:10.3767/003158513X673080.
6. Figure 2 is very good, but difficult to understand due to the large number of similar colors. Perhaps it can be improved somehow? Slightly increase the size of the graph? Make callouts to the columns?
7. The following figures are incorrectly referenced.
line 190 «factor called PacC [36,37] (Fig. 2)» - it should be Fig. 3
line 328 «ATP-dependent manner [83] (Fig. 3)» - it should be Fig. 4
8. Species name error.
lines 363, 369, 376 «Emericellopsis alkaline», it should be Emericellopsis alkalina
After the first full mention of the name of the fungal species, it’s possible to use its abbreviated name (at least in the current section). So, instead of Emericellopsis alkalina, you can write E. alkalina.
9. line 521 «Aspergillus glaucus is an haloalkaliphilic fungi» - fungus
10. The paragraph about "bioactive compounds" (lines 554-560) could be further improved.
The authors have analyzed the genome of S. alkalinus, but it is strange that there is no mention of the works on already isolated bioactive compounds from S. alkalinus and another alkaliphilic fungus Emericellopsis alkalina. For example, Kuvarina et al., 2022; https://doi.org/10.3390/jof8070659.; Rogozhin et al., 2018; DOI: 10.3390/molecules23112785; Kuvarina et al., 2021; DOI: 10.3390/jof7020153
In line 556, at the first mention of the fungus in this section, its name should be written in full. I also consider that in the context of this work, it is necessary to repeat here once again that Sodiomyces alkalinus is an alkaliphilic fungus, despite mentioning it earlier.
5. line 570 «A. sydowii, a case under study» it is necessary to give the full name of the fungus in the title of the section
6. References No.25 is not correct: the title of the article is in capital letters.
7. To the previous version of the manuscript.
No. 68 - the reference is not correct.
It was: 68. Young, B.P.; Shin, J.J.H.; Orij, R.; Chao, J.T.; Li, S.C.; Guan, X.L.; Khong, A.; Jan, E.; Wenk, M.R.; Prinz, W.A.; et al. Phosphatidic Acid Is a pH Biosensor that Links Membrane Biogenesis to Metabolism. Science (80-. ). 2010, 329, 1085–1088.
The output data of the article was incorrect.
Author Response
Dear Reviewer 1,
Thank you very much for reviewing the manuscript. We have incorporated the changes suggested by you and have highlighted the changes in the manuscript.
We attach the file with the point-by-point responses to your suggestions.
Thanks in advance, kind regards.

Reviewer 2 Report
The authors did a good job with the comments. The work can be published in the journal
Author Response
Dear Reviewer 2,
Thank you very much for reviewing the manuscript and for your comments.
Kind regards.